# Restoration of the Joint Line Configuration Reproduces Native Mid-Flexion Biomechanics after Total Knee Arthroplasty: A Matched-Pair Cadaveric Study

**DOI:** 10.3390/bioengineering9100564

**Published:** 2022-10-17

**Authors:** Dai-Soon Kwak, Yong Deok Kim, Nicole Cho, Yong In, Man Soo Kim, Dohyung Lim, In Jun Koh

**Affiliations:** 1Catholic Institute for Applied Anatomy, Department of Anatomy, College of Medicine, The Catholic University of Korea, Seoul 06591, Korea; 2Joint Replacement Center, Eunpyeong St. Mary’s Hospital, Seoul 03312, Korea; 3Department of Orthopaedic Surgery, College of Medicine, The Catholic University of Korea, Seoul 06591, Korea; 4Boston College, Morrissey College of Arts and Sciences, Chestnut Hill, MA 02467, USA; 5Department of Orthopaedic Surgery, Seoul St. Mary’s Hospital, Seoul 06591, Korea; 6Department of Mechanical Engineering, Sejong University, Seoul 05006, Korea

**Keywords:** rollback, ligament strain, kinematic alignment, mechanical alignment, total knee arthroplasty

## Abstract

Background: Recent evidence supports that restoration of the pre-arthritic condition via total knee arthroplasty (TKA) is associated with improved post-TKA performance and patient satisfaction. However, whether the restored pre-arthritic joint line simulates the native mid-flexion biomechanics remains unclear. Objective: We performed a matched-pair cadaveric study to explore whether restoration of the joint line via kinematically aligned (KA) TKA reproduced native knee biomechanics more accurately than the altered joint line associated with mechanically aligned (MA) TKA. Methods: Sixteen fresh-frozen cadaveric knees (eight pairs) were affixed onto a customized knee-squatting simulator for measurement of femoral rollback and medial collateral ligament (MCL) strain during mid-flexion. One knee from each cadaver was randomly designated to the KA TKA group (with the joint line restored to the pre-arthritic condition) and the other to the MA TKA group (with the joint line perpendicular to the mechanical axis). Optical markers were attached to all knees and rollback was analyzed using motion capture cameras. A video extensometer measured real-time variations in MCL strain. The kinematics and MCL strain prior to and following TKA were measured for all specimens. Results: KA TKA was better for restoring the knee kinematics to the native condition than MA TKA. The mid-flexion femoral rollback and axial rotation after KA TKA were consistently comparable to those of the native knee. Meanwhile, those of MA TKA were similar only at ≤40° of flexion. Furthermore, KA TKA better restored the mid-flexion MCL strain to that of the native knee than MA TKA. Over the entire mid-flexion range, the MCL strain of KA TKA and native knees were similar, while the strains of MA TKA knees were more than twice those of native knees at >20° of flexion. Conclusions: The restored joint line after KA TKA effectively reproduced the native mid-flexion rollback and MCL strain, whereas the altered joint line after MA TKA did not. Our findings may explain why patients who undergo KA TKA experience superior outcomes and more natural knee sensations during daily activities than those treated via MA TKA.

## 1. Introduction

Despite advances in technology and surgical technique, recent evidence indicates that mechanically aligned (MA) total knee arthroplasty (TKA) does not improve residual symptoms, natural knee sensations, or patient satisfaction [1,2,3,4,5]. In addition, neutrally aligned TKA fails to reproduce patient-specific knee kinematics [6,7,8]. Thus, kinematic alignment that restores patient-specific pre-arthritic alignment, joint line obliquity, and soft tissue laxity has attracted increasing interest [9,10]. Many studies have shown that kinematically aligned (KA) TKA better restores the pre-arthritic knee kinematics and functional performance than MA TKA, thereby increasing patient satisfaction [11,12,13,14,15,16,17,18]. However, biomechanical data explaining these improvements are lacking.

Many daily activities, including walking and rising from a chair, are performed in the mid-flexion range [19]; restoration of preoperative knee performance within that range is essential for TKA to be successful. It has been suggested that mid-flexion instability is inevitable after well-balanced MA TKA [20]; joint line elevation after MA TKA was a risk factor for instability [21,22,23]. Theoretically, KA TKA that restores both the joint line height and obliquity of the pre-arthritic knee should provide more natural mid-flexion kinematics and laxity than MA TKA. However, reports on the relationship between restoration of the joint line configuration (height and obliquity) and mid-flexion biomechanics/laxity have been inconsistent [12,15,21,24,25,26].

The objective of this matched-pair study was to determine whether the restored pre-arthritic joint line configuration after KA TKA provided femoral rollback closer to that of the native knee than the altered joint line perpendicular to the mechanical axis created after MA TKA, and whether KA TKA more effectively restored MCL strain in comparison to MA TKA. We hypothesized that KA TKA reproduced the natural mid-flexion knee kinematics (rollback and tibiofemoral axial rotation) better than MA TKA. In addition, we proposed that KA TKA would more naturally reproduce MCL strain in the mid-flexion range than MA TKA.

## 2. Materials and Methods

### 2.1. Participants

Eight freshly frozen full-body specimens (human cadavers, donated to the College of Medicine, The Catholic university of Korea, 16 knees; five male pairs and three female pairs; mean age, 76 years; range: 58–86 years) were used (Table 1). The two knees of each cadaver were randomly assigned to either the KA TKA or the MA TKA group. All specimens were macroscopically intact, and none exhibited any gross pathology. This cadaveric study was approved by Institutional Cadaver Research Committee (College of Medicine, The Catholic University of Korea) (R19-A018).

### 2.2. Preparation of Specimens

The specimens were frozen at 20 °C until they were thawed to room temperature on the evening prior to dissection. All skin and subcutaneous tissues were dissected away, leaving only the extensor mechanism, knee capsule, and periarticular soft tissues intact. We took high-resolution anterior-to-posterior photographs of each leg and measured the anatomical and mechanical axes of both the femur and tibia and the hip–knee–ankle axis. The severity of osteoarthritis (OA) in each cadaveric specimen was graded as mild (no articular cartilage lesions), moderate (focal lesion present), or severe (extensive lesion present) (Table 1). Separation of the quadriceps femoris revealed the vastus medialis, rectus femoris/vastus intermedius, and vastus lateralis. Additionally, separation of the hamstring muscles revealed the biceps femoris and semimembranosus/semitendinosus. Then, suturing the parted muscle branches with wire ensured the connection between the muscles. The femur and the tibia were cut 30 cm proximal and 25 cm distal to the joint line, respectively.

### 2.3. Surgical Procedure

A senior surgeon (one of the authors) performed all arthroplasties following a standard posterior-substituting (PS) prosthetic system (Legion Total Knee System; Smith & Nephew, Memphis, TN, USA). Furthermore, a subvastus approach ensured exposure of the knee joint; the patella was not resurfaced in any case. We sought to ensure that all medial (distal and posterior femoral) resections were 9.5 mm in thickness, because the thicknesses of the distal and posterior femoral implants were 9.5 mm. KA TKA was performed using the previously described calipered technique [27,28]. The femur and tibia resection thicknesses were equivalent to those of the implants placed in the native joint lines; there was no manipulation of soft tissue. Calipers were used to measure the thickness of each resected osteochondral fragment, followed by adjustment of each resection until it matched the thickness of the implant (Figure 1). The angle of the tibial resection guide was altered until the saw slot and angle were parallel to the coronal and sagittal proximal articular surfaces (after compensating for wear). In the MA TKA group, TKA was performed with the conventional measured resection technique. Resection of the distal femur proceeded using intramedullary instrumentation that considered the difference between the mechanical and anatomical axes of each individual specimen; the trans-epicondylar axis was used as the reference for determining the femoral component external rotation. Extramedullary instrumentation was then used to perform resections of the coronal and sagittal proximal tibias at a cutting angle of 90° relative to the tibial axis (Figure 1). Lastly, a tensor device (B Braun-Aesculap, Tuttlingen, Germany) under a 200-N distraction force was used to measure the 0° and 90° flexion gaps. The resected osteochondral fragment thickness and gap after bone resection with KA TKA contrasted with those after MA TKA (Table 2).

### 2.4. Test Procedure

Following preparation, each knee was affixed in its original axial position onto a customized knee-squatting simulator system (RNX and Corentec, Seoul, Korea) [18]; this induces continuous flexion–extension knee motion under physiological muscle loading and allows both the femur and the tibia to be positioned with six degrees of freedom (Figure 2A). The ratio between the physiological cross-sectional multiplane loading of the quadriceps and hamstring muscles simulated physiological knee joint loading [29].

#### 2.4.1. Knee Kinematics

A motion capture system combined with optical markers (Cortex 8.1; Motion Analysis, Rohnert Park, CA, USA) was used to measure knee kinematics. Five motion capture cameras (Kestrel 1300; Motion Analysis) were employed. Optical markers were attached to the medial and lateral epicondyles of the femur (the FML and FMM); medial and lateral ends of the longest medial-lateral axis of the tibia (the TBL and TBM); and RIG system (FM1, FM2, and FM3 BTT and BTB) to analyze the medial and lateral femoral rollback (Figure 2B). We used an L-frame and 200-mm wand to calibrate the system. After calibration, the wand length was typically 199.99~200.02 mm. We checked the accuracy during continuous movement by measuring the distance between two markers on the rotational disk. Over 60 s at 60 Hz, the root mean square error was 0.012 mm. Samples fixed to the knee rig system were tested from 20° to 80° of flexion. The marker positions were measured at 60 Hz by the motion capture cameras, and the positional coordinates were calculated (Figure 3A).

#### 2.4.2. MCL Strain

Real-time variations in the mid-flexion MCL strain were examined with a noncontact video extensometer featuring a high-resolution digital camera (ISG Monet 3D; Sobriety s.r.o., Kuřim, Czech Republic) and real-time image processing software (ISG; Mercury RT × 64 2.7; Sobriety). The camera was placed 1 m from each knee (field of view, 485 × 383 mm; resolution, 1.87 µm). In order to minimize the effects of illumination, each test was completed under two 36-W light-emitting diodes. The MCL strain was assessed at knee flexion angles from 20 to 80° at intervals of 10°. The strain was measured over the entire MCL area; measurements were performed on each specimen prior to and following TKA (Figure 3B).

### 2.5. Statistical Analysis

All data were displayed as means with their corresponding standard deviations. We used the paired *t*-test to determine whether the medial and lateral rollback, axial rotation, and MCL strain differed between the preoperative and post-TKA specimens. All analyses were completed using SPSS for Windows software (ver. 26.0; IBM Corp., Armonk, NY, USA), and a *p*-value < 0.05 indicated statistical significance. Additionally, an a priori power analysis based on a pilot test of changes in the femoral rollback and MCL strain of native knees was performed to determine the required sample size. We found that, for two-sided hypothesis testing at an alpha value of 0.05 and power of 90%, seven pairs of knees (14 knees) were needed to detect a 1-mm difference in the rollback and 5% difference in the MCL strain.

## 3. Results

### 3.1. Femoral Rollback

KA TKA restored the mid-flexion medial and lateral rollback and tibiofemoral axial rotation to levels closer to those of the native knee than MA TKA. The medial and lateral rollback of KA TKA and native knees was similar over the entire mid-flexion range (Figure 4A and Figure 5A). The medial and lateral rollback after MA TKA were significantly lower compared with native knees at both > 40° (Figure 4B) and >20° (Figure 5B) of flexion. In addition, tibiofemoral axial rotation during flexion after KA TKA was similar to that of the native knee (Figure 6A), while that of MA TKA differed from the native knee in the mid-flexion range (Figure 6B). Remarkably, the femur moved forward during flexion after MA TKA over the entire mid-flexion range, except at 20° of flexion (Figure 4B, Figure 5B and Figure 6B).

### 3.2. MCL Strain

KA TKA was better for restoring the MCL strain to that of the native knee over the entire mid-flexion range than MA TKA. The mean strain measurements following KA TKA and those of the native knee were alike over all ranges (Figure 7A). The MCL strain after MA TKA was two-fold greater than that of the native knee at flexion angles > 20° (Figure 7B).

## 4. Discussion

Despite advancements in both technology and surgical techniques of MA TKA, patient dissatisfaction with post-TKA pain relief and overall outcomes remains high; a substantial proportion of patients report knee abnormalities [1,2,5,30,31]. KA TKA seeks to restore the anatomy of each individual patient; the kinematic and clinical outcomes are better than those of MA TKA [14,15,16,18]. Joint line elevation after MA TKA is associated with a risk of mid-flexion instability; theoretically, KA TKA restoration of native joint line height and obliquity makes mid-flexion biomechanics more natural than MA TKA [20,21,25,32]. However, it remains unclear whether restoration of the joint line configuration affects post-TKA mid-flexion kinematics and laxity [12,18,21,24,25,26]. Therefore, this matched pair cadaveric study tested which TKA alignment concept, KA or MA, would reproduce more native mid-flexion rollback and MCL strain.

The present study’s results endorse that KA TKA provides better physiological kinematics over the mid-flexion range than MA TKA, as hypothesized. We found that after KA TKA, medial and lateral femoral rollback, and axial rotation, were consistently similar to those of the native knee, whereas for MA TKA this was the case only at ≤40° of flexion. In addition, a paradoxical femoral forward movement during flexion was observed after MA TKA. Our findings agree with those of recent cadaveric studies, which suggest that compared to MA TKA, knee kinematics after KA TKA were more alike to those of native knees [14,15,26]. Our results, and those of previous studies, reveal why patients who undergo KA TKA often report superior mid-flexion functional performance compared with those who undergo MA TKA.

Our findings also support the hypothesis that the joint line following KA TKA is better for restoring natural MCL strain during mid-flexion than the perpendicular joint line after MA TKA. KA TKA resulted in MCL strain that was consistently comparable to that of native knees, whereas, in the case of MA TKA, it was twice as high. Our findings agree with those of a recent cadaveric study: KA TKA was better for restoring the magnitude and distribution of MCL strain to natural levels than MA TKA [18]. Although it is challenging to directly liken our results with those of prior studies that assessed MCL strain via linear, two-dimensional measurements of length changes under valgus stress, our findings support previous studies that report that restoration of the pre-arthritic joint line provided a more physiological MCL strain than traditional MA TKA [12,14,15,26,33]. Furthermore, our results, when taken into consideration with the widely acknowledged existence of a nociceptor in the MCL [34], indicate that patients may experience less pain and more native knee sensations during mid-flexion after KA TKA, as opposed to MA TKA.

The results of the present study propose that restoration of natural MCL strain via KA TKA may explain the more physiological knee kinematics evident after KA TKA compared to MA TKA. In this study, KA TKA reproduces more physiological medial pivot motion, while MA TKA results in paradoxical anterior motion during mid-flexion. Given that the MCL serves as the fundamental restraint in ACL-deficient, prosthetic knees, our findings suggest that MCL strain may be strongly associated with restoration of knee kinematics. Interestingly, a recent cadaveric study reported that restoration of joint line obliquity was not associated with mid-flexion coronal plane laxity, if the medial joint line height was restored [21]. However, although we restored the medial joint line height in all knees, our study found a significant difference between KA and MA TKA knees in the MCL strain. Still, the correlation between these findings is limited, since the previous study assessed soft tissue laxity by the length changes of MCL after the valgus load, and information on knee kinematics was not presented. Although the cause of the inconsistencies is unclear, one plausible explanation is that, although valgus stress increases the MCL strain, the MCL length may not change if the stress is lower than the threshold required for such change.

Due to the use of a cadaveric model, this study has some limitations. First, specimen preparation and the squatting loads used may not have been entirely natural. Second, as the tissue quality around the knee joint is associated with the severity of knee OA, the experimental results could be affected by the OA status of the cadaver. However, most knees in this study lacked advanced osteoarthritis necessitating TKA, so caution is required when extrapolating our findings to clinical practice. Third, this study featured PS prosthesis, which requires consideration of the implant feature prior to any broad generalizations, as implant design has been shown to be strongly correlated with knee kinematics [26,35]. Generally, when performing KA TKA, a cruciate-retaining (CR) prosthesis is recommended. However, a recent study found that native knees, and knees in which CR and PS prostheses were placed during KA TKA, had similar kinematics and soft tissue laxity [14]. Fourth, because the thresholds for pain and mechanical failure in the human knee are unknown, assessing the clinical significance of MCL strain was difficult. Fifth, it is possible that the study was underpowered and subject type-II error with respect to detecting all relevant outcomes. Sixth, the group assignment for the knees were not kept blinded to all investigators, which could have raised ascertainment bias. Finally, biofeedback, which is natural in knee function and enhanced by the innovative TKA surgical technique, could have introduced nonlinearity in the feedback and improved the signal-to-noise ratio in the loop [36]. Thus, the video extensometer that we used does not process data with 100% accuracy; changes in illumination may affect image processing, because illumination affects the properties of biomaterials. Nevertheless, we used a matched-pair design to minimize confounders, and this is the first study to simultaneously measure femoral rollback and MCL strain. These results provide valuable insight on the differences regarding mid-flexion kinematics and patterns of MCL strain between KA TKA and MA TKA.

## 5. Conclusions

We investigated whether restoration of the pre-arthritic joint line following TKA would affect post-TKA biomechanics. Restoration of the height and obliquity of the pre-arthritic joint line following KA TKA reproduces more natural rollback and MCL strain than alteration of the joint line following MA TKA over the entire mid-flexion range. Future studies focused on the development of both the motion analysis system that assesses the knee kinematics of patients in real clinical practice and the algorithm that recommends the optimal implant position restoring native knee kinematics are required.

## Figures and Tables

**Figure 1 bioengineering-09-00564-f001:**
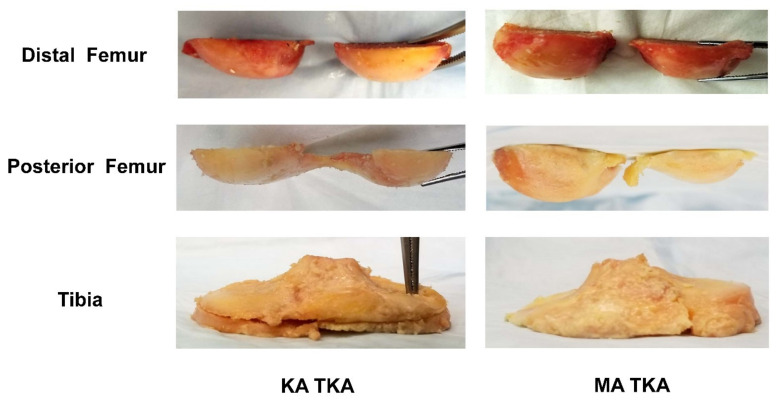
The resected osteochondral fragments. During KA TKA, the femur and tibia resection thicknesses were the same as those of the implants placed in the native joint lines. In the MA TKA group, distal femoral resection was performed perpendicular to the mechanical axis of the femur; the trans-epicondylar axis served as the reference when determining the external rotation of the femoral component. Tibial resection was then performed perpendicular to the mechanical axis of the tibia.

**Figure 2 bioengineering-09-00564-f002:**
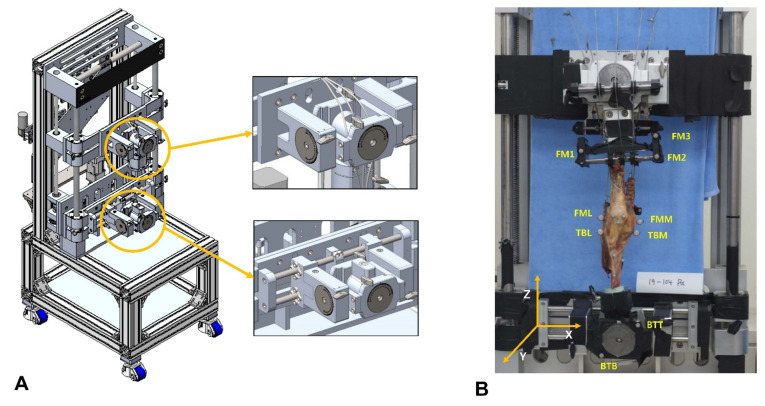
Schematic of the knee-squatting simulator with six degrees-of-freedom (**A**) and locations of the optical markers (**B**) on the femur (FML and FMM); tibia (TBL and TBM); and RIG system (FM1, FM2, and FM3 BTT and BTB).

**Figure 3 bioengineering-09-00564-f003:**
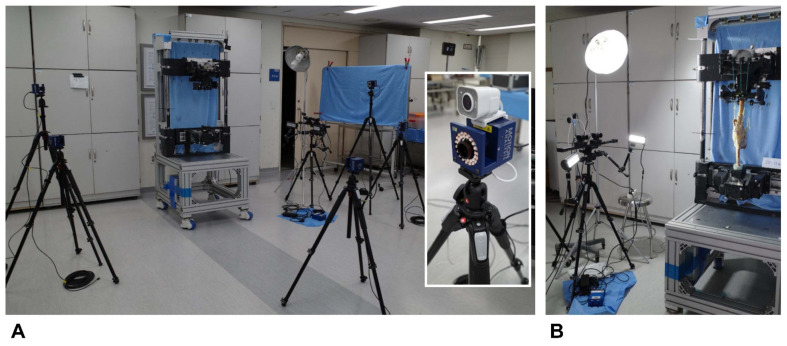
Biomechanical test setup. The motion capture system included five cameras; optical markers were placed when measuring knee kinematics (**A**). Real-time changes in MCL strain during flexion were analyzed using a noncontact video extensometer with a high-resolution digital camera and real-time image-processing software. A camera was placed 1 m from each specimen mounted in a customized knee-squatting simulator (**B**).

**Figure 4 bioengineering-09-00564-f004:**
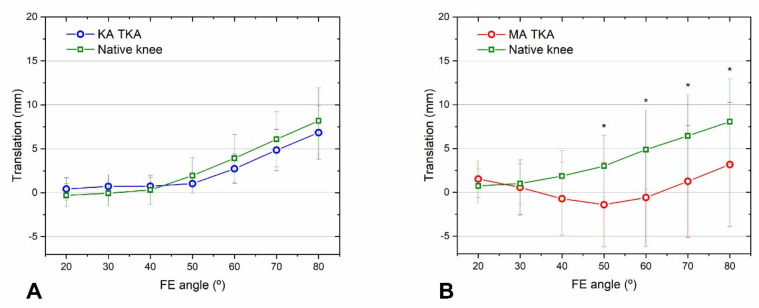
Medial femoral rollback after KA TKA (**A**) and MA TKA (**B**). Rollback following KA TKA and of the native knees were similar at all flexion angles. Meanwhile, rollback following MA TKA was reduced at flexion angles > 40°. Error bars denote standard deviations. Significant differences (*p* < 0.05) are marked with asterisks.

**Figure 5 bioengineering-09-00564-f005:**
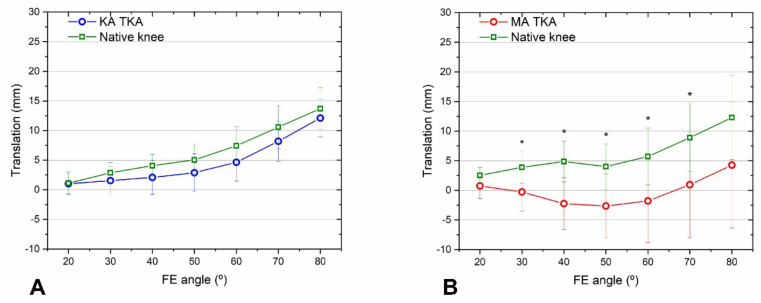
Lateral femoral rollback after KA TKA (**A**) and MA TKA (**B**). Rollback after KA TKA compared to native knee rollback over the entire mid-flexion range, but in the case of MA TKA, the rollback was significantly smaller, except at a flexion angle of 20°. A paradoxical forward movement was observed after MA TKA. Error bars denote standard deviations. Significant differences (*p* < 0.05) are marked with asterisks.

**Figure 6 bioengineering-09-00564-f006:**
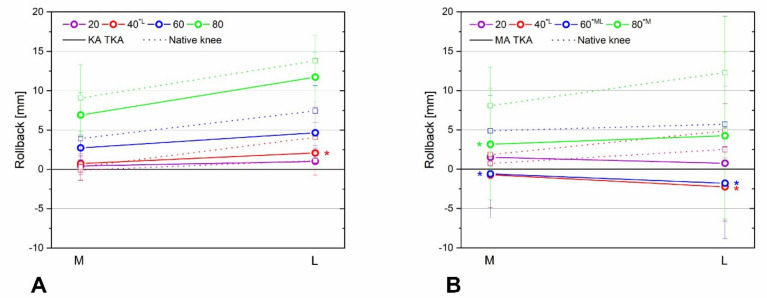
Tibiofemoral axial rotations after KA TKA (**A**) and MA TKA (**B**). During flexion, KA TKA and native knee rotations were similar, while rotation following MA TKA was quite different from that of the native knee. A paradoxical forward movement was observed after MA TKA. Significant differences (*p* < 0.05) are marked with asterisks.

**Figure 7 bioengineering-09-00564-f007:**
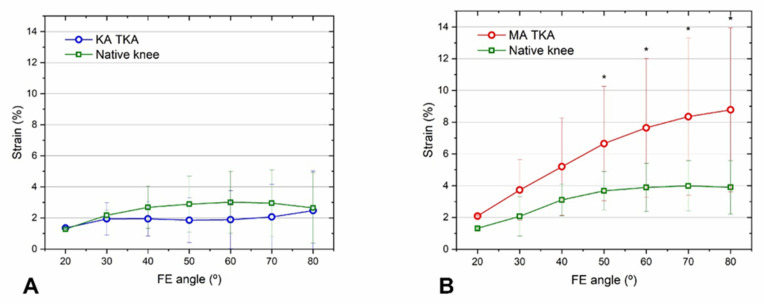
MCL strain after KA TKA (**A**) and MA TKA (**B**). The mean strains following KA TKA and that of the native knee were consistently similar, while the mean strain after MA TKA was about two-fold higher at >30° of flexion. Error bars denote standard deviations. Significant differences (*p* < 0.05) are marked with asterisks.

**Table 1 bioengineering-09-00564-t001:** Demographic variables of the specimens.

Specimen Number	Age (y)	Hight (cm)	Weight (kg)	Sex	Left Knee HKA (°)	Right Knee Alignment	OA Severity	HKA (°)	Alignment	OA Severity
1	85	163	74	Male	3.6	MA	Mild	5.2	KA	Mild
2	71	175	64	Male	2.9	KA	Mild	2.9	MA	Mild
3	84	163	50	Male	7.1	MA	Mild	8.0	KA	Mild
4	58	166	78	Male	5.1	MA	Mild	2.1	KA	Mild
5	86	167	48	Female	4.0	MA	Mild	0	KA	Mild
6	79	164	58	Female	3.3	KA	Moderate	4.4	MA	Moderate
7	63	170	56	Male	7.6	MA	Mild	4.6	KA	Mild
8	81	160	48	Female	3.5	KA	Mild	1.0	MA	Mild

**Table 2 bioengineering-09-00564-t002:** Resected osteochondral fragment thickness and gaps after KA and MA TKA.

	KA TKA (*n* = 8)	MA TKA (*n* = 8)	Significance
Resected bone thickness (mm)			
Distal femur			
Medial	9.6 (0.7)	10.1 (0.8)	0.227
Lateral	9.8 (0.7)	7.0 (1.1)	<0.001
Posterior femur			
Medial	10.4 (1.5)	11.5 (1.5)	0.158
Lateral	10.5 (1.6)	8.3 (1.0)	<0.001
Tibia			
Medial	6.3 (1.5)	2.1 (0.3)	<0.001
Lateral	6.9 (0.8)	9.6 (1.7)	<0.001
Gap (mm)			
Full extension			
Medial	11.1 (2.2)	12.8 (1.0)	0.076
Lateral	11.8 (2.4)	12.8 (1.0)	0.293
90° flexion			
Medial	12.8 (2.4)	13.4 (1.2)	0.525
Lateral	16.0 (2.3)	13.9 (1.3)	0.078

All data are the mean (standard deviation). KA, kinematic alignment; MA, mechanical alignment; TKA, total knee arthroplasty.

## Data Availability

All data are presented in the article. Instrumental readings are available upon request from the corresponding author.

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
