# Peer review of "Restoration of the Joint Line Configuration Reproduces Native Mid-Flexion Biomechanics after Total Knee Arthroplasty: A Matched-Pair Cadaveric Study"

_bioengineering, 2022, doi:10.3390/bioengineering9100564_

Round 1

Reviewer 1 Report

This paper performed a match-pair study to investigate the difference between the restoration of the joint line by KA TKA and the altered joint line by MA TKA. The experiments were tested with 16 cadaveric knees. The authors concluded that the KA TKA was better than the MA TKA in restoration of knee kinematics and medial collateral ligament strain during mid-flexion. The following comments could be considered for improvement:

1) In Table 1, it is noted that most of the knees were from elderly cadavers (age > 60 years). It could be better to discuss whether the experimental results could be affected by aging.

2) It is a bit confused about the ethics statement. The "Institutional Review Board Statement: This cadaveric study was exempt from the institutional re- 328 view board of our institution because it did not involve human subjects." In Section 2.1, the text states "Eight freshly frozen full-body specimens (16 knees; five male pairs and three female 81 pairs; mean age, 76 years; range: 58–86 years) were used." The specimens seems not human subjects, but which animals were used? The source of the experiment specimens should be presented in the revised manuscript.

3) In Conclusion section, the major contribution and key findings in the study should be addressed, instead of the ßbackground descriptions.

4) The "keywords: keyword1; keyword2;..." should be removed in the Abstract section.

Author Response

Reviewer 1

This paper performed a match-pair study to investigate the difference between the restoration of the joint line by KA TKA and the altered joint line by MA TKA. The experiments were tested with 16 cadaveric knees. The authors concluded that the KA TKA was better than the MA TKA in restoration of knee kinematics and medial collateral ligament strain during mid-flexion. The following comments could be considered for improvement:

â–¶The authors appreciate the reviewer’s understanding and succinct summary of our study.

1) In Table 1, it is noted that most of the knees were from elderly cadavers (age > 60 years). It could be better to discuss whether the experimental results could be affected by aging.

â–¶The authors are aware of the reviewer's concerns and agree with the reviewer's point. In general, as knee kinematics are determined by articular geometry and soft tissue status, the experimental results could have been affected by various factors. In real clinical practice, we infrequently encountered young patients with severe OA and elderly patients with mild OA. We also found that the anatomical condition of articular surface geometry and soft tissue are more strongly associated with the severity of knee osteoarthritis (OA), rather than age itself. Thus, we classified the severity of cadaveric knee OA in this study into mild, moderate and severe (Lines 95 -97), and added this information to the revised Table 1. Although the mean cadaveric age was relatively high, all knees in this study showed no articular cartilage lesions, except one pair showing focal articular cartilage defect. As we described, in the Discussion section of our original manuscript, most specimens in this study lacked advanced OA necessitating TKA. Thus, it is difficult to determine whether the degeneration of tissue around the knee joint would have affected our results. We have now addressed this issue in the Limitation section (Line 242-246). We hope that these revisions address the reviewer’s concerns satisfactorily.

2) It is a bit confused about the ethics statement. The "Institutional Review Board Statement: This cadaveric study was exempt from the institutional re- 328 view board of our institution because it did not involve human subjects." In Section 2.1, the text states "Eight freshly frozen full-body specimens (16 knees; five male pairs and three female 81 pairs; mean age, 76 years; range: 58–86 years) were used." The specimens seem not human subjects, but which animals were used? The source of the experiment specimens should be presented in the revised manuscript.

â–¶The authors recognize the reviewer’s concern and agree with the excellent point. As the cadaveric specimen is neither a human nor an animal, this cadaveric test was exempted from the IRB of our institution. However, the independent committee, Institutional Cadaver Research Committee (College of Medicine, The Catholic University of Korea), supervises and approves human cadaveric tests at our institute. We rewrote Section 2.1 (Lines 82-88) and the IRB statement (Lines 288-291) for better understanding. We hope that these revisions address the reviewer’s concerns satisfactorily.  

3) In Conclusion section, the major contribution and key findings in the study should be addressed, instead of the background descriptions.

 â–¶The authors are aware of the reviewer's concerns and agree with the reviewer's point. As suggested, we rewrote the Conclusion section (Lines 266-272). We hope that these revisions address the reviewer’s concerns satisfactorily.

4) The "keywords: keyword1; keyword2;..." should be removed in the Abstract section.

â–¶As suggested, we removed the keywords in the Abstract section.

Reviewer 2 Report

The paper regard important results that are related to bioengineering mechanics and could involve important issues for the future. The paper is well posed and both the results and the experimental apparatus are well posed. The intersting results lead me to give an enginerring and control reason that is justified by theory. Bio feedback that is natural in the knee funcion and that is improved by the innovative orthopedic surgery method introduced could introduce nonlinearity in the feedback that could improve the noise and the disturbance in the loop. This could justify the results. It is therefore suggested to include the following contribution:

Can Noise in the Feedback Improve the Performance of a Control System? Autorhors Maide Bucolo, Arturo Buscarino, Luigi Fortuna, Salvina Gagliano Date e 2021/7/15 Journal of the Physical Society of Japan Volume 9 0 Numebr 7 Pagies 075002 Editor The Physical Society of Japan THe previous consideration could open a new research area and could make the results more appealing for  wider range of readers. It is therefore that I suggest the authors to spend some words about.

Author Response

The paper regard important results that are related to bioengineering mechanics and could involve important issues for the future. The paper is well posed and both the results and the experimental apparatus are well posed. The interesting results lead me to give an engineering and control reason that is justified by theory. Bio feedback that is natural in the knee function and that is improved by the innovative orthopedic surgery method introduced could introduce nonlinearity in the feedback that could improve the noise and the disturbance in the loop. This could justify the results. It is therefore suggested to include the following contribution:

Can Noise in the Feedback Improve the Performance of a Control System? Autorhors Maide Bucolo, Arturo Buscarino, Luigi Fortuna, Salvina Gagliano Date e 2021/7/15 Journal of the Physical Society of Japan Volume 9 0 Numebr 7 Pagies 075002 Editor The Physical Society of Japan The previous consideration could open a new research area and could make the results more appealing for a wider range of readers. It is therefore that I suggest the authors to spend some words about.

â–¶The authors agree with the points nicely stated by reviewer. As suggested, we addressed this issue in the Discussion section (Lines 256-258) and cited a new reference [Ref 36]. We hope that these revisions address the reviewer’s concerns satisfactorily.

Reviewer 3 Report

The article entitled “Restoration of the joint line configuration reproduces native mid-flexion biomechanics after total knee arthroplasty: A matched-pair cadaveric study”. The aim of this matched-pair cadaveric study was investigated whether the restored pre-arthritic joint line configuration after KA TKA provided femoral rollback closer to that of the native knee than the altered joint line perpendicular to the mechanical axis created after MA TKA, and whether KA TKA more effectively restored MCL strain in comparison to MA TKA.

Below are some suggestions: 

In the Abstract:

- The abstract is well written, in a structured way, I suggest just highlighting the objective of the research.

In the Introduction:

- The introduction can be improved, highlighting the clinical importance of the research, as it was written based on results found in previous research that should be in the discussion and not in the introduction. I also suggest ending the introduction by setting the objective.

In the Materials and Methods:

The methodology is well described. I suggest putting the origin of the cadaver: Do you have approval from any institutional body?

In the Results:

The results are well presented, I suggest, if possible, to improve the quality of the figures.

One question: the average age of the species used was 58-86 years, do the authors believe that the results would be different if younger cadavers were used? 

In the Discussion:

The discussion is written correctly, comparing the results of this study with findings from the literature, as well as bringing the limitations of the study. I suggest only inserting the future perspectives of the group. 

In the Conclusion:

I suggest inserting a conclusion with final considerations, including the purpose of the research, its main results and future clinical perspectives.

Author Response

Below are some suggestions: 

In the Abstract:

- The abstract is well written, in a structured way, I suggest just highlighting the objective of the research.

â–¶The authors agree with the points nicely stated by reviewer. As suggested, we rewrote the sentence. (Lines 22)

In the Introduction:

- The introduction can be improved, highlighting the clinical importance of the research, as it was written based on results found in previous research that should be in the discussion and not in the introduction. I also suggest ending the introduction by setting the objective.

â–¶The authors agree with the points nicely stated by reviewer. As suggested, we removed the results of previous studies and rewrote the Introduction section (Lines 71-79)

In the Materials and Methods:

The methodology is well describedI suggest putting the origin of the cadaver: Do you have approval from any institutional body?

â–¶The authors recognize the reviewer’s concern and agree with the excellent point. As the cadaveric specimen is neither a human nor an animal, this cadaveric test was exempted from the IRB of our institution. However, the independent committee, Institutional Cadaver Research Committee (College of Medicine, The Catholic University of Korea), supervises and approves human cadaveric tests at our institute. We rewrote Section 2.1 (Lines 82-88) and the IRB statement (Lines 288-291) for better understanding. We hope that these revisions address the reviewer’s concerns satisfactorily.  

In the Results:

The results are well presented, I suggest, if possible, to improve the quality of the figures.

One question: the average age of the species used was 58-86 years, do the authors believe that the results would be different if younger cadavers were used? 

â–¶The authors are aware of the reviewer's concerns and agree with the reviewer's point. We improved the resolution of the figure to 600 dpi. On the other hand, as knee kinematics are determined by articular geometry and soft tissue status, the experimental results could have been affected by various factors. In real clinical practice, we infrequently encountered young patients with severe OA and elderly patients with mild OA. We also found that the anatomical condition of articular surface geometry and soft tissue are more strongly associated with the severity of knee osteoarthritis (OA), rather than age itself. Thus, we classified the severity of cadaveric knee OA in this study into mild, moderate and severe (Lines 95 -97), and added this information to the revised Table 1. Although the mean cadaveric age was relatively high, all knees in this study showed no articular cartilage lesions, except one pair showing focal articular cartilage defect. As we described, in the Discussion section of our original manuscript, most specimens in this study lacked advanced OA necessitating TKA. Thus, it is difficult to determine whether the degeneration of tissue around the knee joint would have affected our results. We have now addressed this issue in the Limitation section (Line 242-246). We hope that these revisions address the reviewer’s concerns satisfactorily.

In the Discussion:

The discussion is written correctly, comparing the results of this study with findings from the literature, as well as bringing the limitations of the study. I suggest only inserting the future perspectives of the group. 

â–¶The authors appreciate the reviewer’s comments. As suggested, we added future perspectives in the Conclusion section (Lines 266-272).

In the Conclusion:

I suggest inserting a conclusion with final considerations, including the purpose of the research, its main results and future clinical perspectives.

â–¶The authors agree with the points nicely stated by reviewer. As suggested, we rewrote the Conclusion section (Lines 266-272).

Reviewer 4 Report

1. How was it ensured that the cadavers did not have any knee joint abnormality before? Particularly, degenerative knee disorders. It is very much likely considering the age group (maximum was 86 years).

2. How was the adequacy of sample size (8 each) can be justified?

3. Blinding during the assessment could have helped in reducing the ascertainment bias. This is a limitation of the study.

Author Response

  1. How was it ensured that the cadavers did not have any knee joint abnormality before? Particularly, degenerative knee disorders. It is very much likely considering the age group (maximum was 86 years).

â–¶The authors are aware of the reviewer's concerns and agree with the reviewer's point. In general, as knee kinematics are determined by articular geometry and soft tissue status, the experimental results could have been affected by various factors. In real clinical practice, we infrequently encountered young patients with severe OA and elderly patients with mild OA. We also found that the anatomical condition of articular surface geometry and soft tissue are more strongly associated with the severity of knee osteoarthritis (OA), rather than age itself. Thus, we classified the severity of cadaveric knee OA in this study into mild, moderate and severe (Lines 95 -97), and added this information to the revised Table 1. Although the mean cadaveric age was relatively high, all knees in this study showed no articular cartilage lesions, except one pair showing focal articular cartilage defect. As we described, in the Discussion section of our original manuscript, most specimens in this study lacked advanced OA necessitating TKA. Thus, it is difficult to determine whether the degeneration of tissue around the knee joint would have affected our results. We have now addressed this issue in the Limitation section (Line 242-246). We hope that these revisions address the reviewer’s concerns satisfactorily.

  1. How was the adequacy of sample size (8 each) can be justified?

â–¶The authors recognize the reviewer’s concern regarding the sample size. Usually, as the sample size of a t test is principally determined by Type I error probability (α), power (β), difference in means (δ), and within group standard deviation (σ), it may be presented differently based on the characteristics of the assessed specimen and the difference detected. Our sample size estimation was based on our three previous studies regarding post-TKA biomechanics [ Koh et al. Orthop Traumatol Surg Res. 2019;105:605-611, Koh et al. Arch Orthop Trauma Surg 2021;141;119-127, Lim et al. Knee Surg Sports Traumatol Arthrosc 2022:30;2815-2823]. Seven knees in each group were required for a power of 0.90 at an alpha level of 0.05 for detecting a 1 mm difference of femoral rollback and 5% difference in MCL strain using a two-sample t test. However, as we mentioned above, the sample size can be affected by the characteristics of study population and the value of minimal clinically important difference (MCID), so it is possible that the study was underpowered and subject to type-II error with respect to detecting all relevant outcomes. We have addressed this issue in the Limitation section (Lines 253-256). We hope that these revisions address the reviewer’s concerns satisfactorily.

  1. Blinding during the assessment could have helped in reducing the ascertainment bias. This is a limitation of the study.

â–¶ The authors are aware of the reviewer's concerns and agree with the reviewer's point. We agree that keeping the group assignment blind to all investigators would reduce the ascertainment bias. We added this issue to the Limitation section (Lines 255-256). We hope that these revisions address the reviewer’s concerns satisfactorily.

Round 2

Reviewer 4 Report

nil